# Lupin Kernel Fibre: Nutritional Composition, Processing Methods, Physicochemical Properties, Consumer Acceptability and Health Effects of Its Enriched Products

**DOI:** 10.3390/nu14142845

**Published:** 2022-07-11

**Authors:** Rahil Malekipoor, Stuart K. Johnson, Rewati R. Bhattarai

**Affiliations:** 1School of Molecular and Life Sciences, Faculty of Science and Engineering, Curtin University, Bentley, WA 6102, Australia; rahil.malekipoor@postgrad.curtin.edu.au (R.M.); stuart@ingredientsbydesign.com.au (S.K.J.); 2Ingredients by Design Pty Ltd., Lesmurdie, WA 6076, Australia

**Keywords:** Lupin, processing, lupin kernel fibre, food ingredient, composition, techno-functionality, health benefits, consumer acceptability

## Abstract

The kernels (dehulled seeds) of lupins (Lupinus spp.) contain far higher dietary fibre levels than other legumes. This fibre is a complex mixture of non-starch polysaccharides making up the thickened cell walls of the kernel. The fibre has properties of both insoluble and soluble fibres. It is a major by-product of the manufacture of lupin protein isolates, which can be dried to produce a purified fibre food ingredient. Such an ingredient possesses a neutral odour and flavour, a smooth texture, and high water-binding and oil-binding properties. These properties allow its incorporation into foods with minimum reduction in their acceptability. The lupin kernel fibre (LKF) has demonstrated beneficial effects in clinical studies on biomarkers for metabolic diseases such as obesity, type 2 diabetes, and cardiovascular disease. It can be described as a “prebiotic fibre” since it improves gut micro-floral balance and the chemical environment within the colon. Thus, LKF is a health-functional ingredient with great opportunity for more widespread use in foods; however, it is evident that more non-thermal methods for the manufacture of lupin kernel fibre should be explored, including their effects on the physicochemical properties of the fibre and the effect on health outcomes in long term clinical trials.

## 1. Introduction

The genus Lupinus consists of hundreds of species of legumes, but only a few, including the white lupin (*Lupinus albus*), the narrow-leafed or Australian sweet lupin (*L. angustifolius*), the yellow lupin (*L. luteus*), and the Andean lupin (*L. mutabilis*) are domesticated for seed production. Western Australia harvests most of the world’s lupin seed (from the low alkaloid Australian sweet lupin) which is considered to be an important nitrogen-fixing rotation crop favouring sustainable cereal production. Australia’s lupin seed is traditionally used for animal feed; however, current interest has shifted towards using lupin seed fractions such as flour protein concentrates/isolates and dietary fibre as human food (Figure 1). This interest may, in part, be due to the high protein and fibre content of lupin flour and the increasing body of evidence from human clinical studies of its potential to reduce the cluster of risk factors that make up metabolic syndrome. In addition, being a gluten-free, non-genetically modified option as well as low in phytoestrogen, lupin may enhance high consumer appeal.

The whole lupin seed contains very high levels of dietary fibre in comparison with those of other legumes, including soybean [1], with as much as 50 g/100 g dry basis (db).

In lupin seeds, the dietary fibre is located in the protective seed coat (hull) at 80–87 g/100 g db [2], of which half (approx. 40 g/100 g db) is contained within the kernel (endosperm) [3]. This lupin kernel fibre (LKF), in the form of thickened cell walls that serve as storage of carbohydrates for seed germination, was first investigated for its potential as a healthy food ingredient several decades ago, and now its potential commercial value is once again being recognised.

This literature review will focus on the current public domain knowledge of the composition, manufacturing methods, and techno-functional as well as health-functional properties of LKF. The drivers and barriers to its potential commercialisation as a food ingredient will also be discussed.

## 2. Dietary Fibre in Lupin Seeds

Lupin seeds contain both insoluble and soluble dietary fibres. The types of dietary fibres have different chemical structures, especially the level of cellulose vs. non-cellulosic polysaccharides, different physicochemical properties (e.g., water binding, water solubility and viscosity), and different effects on the gut (e.g., fermentability) and metabolism (i.e., health benefits) [4,5,6]. Insoluble fibres are not water-soluble nor fermented to any significant extent in the human colon; they primarily consist of cellulose, hemicelluloses, and lignin. In contrast, soluble fibres are highly water-soluble and highly fermentable in the colon, while including pectins, gums, and mucilages [7]. Regarding fibre properties and glycemic regulation, Goff et al. [8] reported that, while the viscosity of soluble fibres is well regarded to regulate glucose metabolism, insoluble fibres, despite limited viscosity, lead to improved glycemic control through four different mechanisms—delay in gastric emptying, gut hormonal regulation, reduced activity of small intestinal digestive enzymes, and delayed absorption of sugars.

There are many excellent and comprehensive reviews on the health benefits of dietary fibre for humans, e.g., Anderson et al. [9,10,11]. In general, insoluble fibres can provide health benefits to the human gastrointestinal tract mainly through their bulking ability which can stimulate healthy bowel movements [10]. However, soluble fibres are generally considered to have more beneficial metabolic effects than those from insoluble fibres [9]. For example, their viscosity can help modulate nutrient digestion and absorption (e.g., blood glucose control). On the other hand, their fermentability can both (a) assist in maintaining a healthy gut microflora and (b) produce a range of metabolically beneficial fermentation products such as short-chain fatty acids that assist with cholesterol control and provide a substrate for healthy colon cell development [10,11].

### 2.1. Dietary Fibre of Lupin Hull

The seed coat (hull) of lupin represents about 25% of the weight of the seed and consists mainly of structural non-starch polysaccharides that are classified as insoluble dietary fibres combined with low levels of protein and lipids, minerals, and phytochemicals such as polyphenols [12,13]. The non-starch polysaccharide in the hull is primarily cellulose with only low levels of lignins (an anti-nutritional factor) [14]. This hull dietary fibre has been characterised as 96.5% insoluble dietary fibre and 3.5% soluble dietary fibre. Recently, attempts to increase the level of soluble dietary fibre in lupin hull using extrusion cooking reported that a slight increase could be achieved [12]. The lupin seed hull has been used as an ingredient in human food, such as in high-fibre bread and meat products, and as a bulking agent [13]; however, most lupin hull undergoes little if any value addition and is disposed of as waste.

The remainder of this literature review will now focus on LKF.

### 2.2. Dietary Fibre of Lupin Kernel

In contrast to the hull, the fibre in the lupin kernel contains more of the soluble fraction and a wider range of different classes of polysaccharides, including pectin substances, cellulose, and non-starch non-cellulosic glucans, with an absence of lignin [14]. However, the proportion of insoluble vs. soluble fibre varies widely in the literature. For instance, Naumann et al. [15] reported that the dietary fibre fraction of dehulled lupin seeds was primarily (approx. 90%) insoluble. In contrast, Turnbull et al. [16] found equal levels of insoluble and soluble dietary fibre in a purified LKF ingredient (88 g/100 g as its total dietary fibre). A reason for this could be the form in which the fibre was analysed, i.e., directly using lupin kernel flour for the fibre assay or firstly isolating the fibre before assaying its dietary fibre composition, which may have modified the cell wall construction making the constituent polysaccharides more soluble. In light of this, it is important to directly analyse the insoluble vs. soluble ratio of any lupin kernel dietary fibre ingredient processed in new or modified ways.

The dietary fibre in the kernels is in the form of thickened walls of the mesophyll cells [17,18], which mainly consist of non-starch polysaccharides and raffinose family oligosaccharides such as raffinose, stachyose, and verbascose [19,20]. More specifically, it is composed of the monosaccharides galactose (67.6%), arabinose (11.5%), uronic acids (8.1%), glucose (7.6%), and xylose (2.6%) [15]. The structure of the lupin kernel cell wall fibre is described as “*1-4-linked long-chain galactans and highly branched 1-5-linked arabinans, which are linked to the rhamnosyl residues of a rhamnogalacturonan backbone*” [21]. The presence of galacturonic acid backbone in the primarily insoluble cell wall fibre means that its polysaccharides structure is more similar to pectin, a soluble fibre, than to commonly found insoluble cellulosic dietary fibres [17]. Therefore, upon processing, we hypothesise that the “trapped” pectin within the cell wall changes from insoluble to being released from the cell wall matrix and thus becoming soluble.

## 3. Manufacture of LKF as a Food Ingredient

### 3.1. Processing Approaches

Lupins, owing to their high protein and dietary fibre contents, have great potential in the manufacture of plant-based food ingredients, which are currently in great demand by the food industry. However, in manufacturing lupin protein concentrates/isolates, the LKF fraction is the main by-product for which value-added commercial utilisation is still in its infancy. The aim of any fractionation is to isolate and quantify fractions of interest and eliminate unwanted components [22]. Potential methods to manufacture LKF can be split into two most common processing types; wet (chemical) processing and dry (physical) processing [23], both to separate the protein and lipid from the dietary fibre using the starting material of lupin kernel flour, flakes, or grits. In addition, enzymatic extractions or combined extraction methods can be used [24,25]. The processing method can greatly affect the composition and properties of the resultant fibre-enriched fraction in food applications and its effect on the human body [25]. This literature review will focus on a comparison between dry and wet processing approaches.

#### 3.1.1. Wet Processing: Methods, Advantages and Disadvantages

The alkaline extraction-isoelectric precipitation method is the most common wet processing technique reported for the separation of protein and fibre of lupin kernels [26]. An example of a wet processing scenario to produce highly dietary-fibre enriched LKF is presented in Figure 2 in which the wet fibre residue is a major by-product of protein concentrate production. First, the protein is extracted from the wet-milled kernel or lupin flour at a high alkaline pH. Centrifugation then results in the protein extract (which is further processed to produce the protein concentrate/isolate) and the fibre residue. This approach provides large volumes of this high-moisture (approx. 80%) paste-like residue that has proven difficult to economically dry to a shelf-stable powder due in part to its very high water-binding properties. Spray drying or freeze-drying to produce the final dry powder ingredient has been reported in the literature [15,27,28]. However, innovation and optimisation for a commercially viable drying method are still required.

There are some drawbacks associated with wet processing for the manufacture of the lupin fractions. The major drawback of this fractionation approach is the requirement of large quantities of water, energy, and chemicals [29]. The high costs of wet processing are also due to extensive losses of solids in the acid-soluble whey (Figure 2) and the need to dry the products as well as recycle the effluents [30]. This method is also time-consuming [25]. In addition, the alkaline extraction and drying steps in wet processing may negatively impact important physico-functional properties of the fibre [26,28].

The techno-functional properties such as solubility and chemical composition of lupin fibre are affected by the extraction conditions; therefore, optimising the condition is crucial [24] to, for instance, improve the fibre yield and maintain or enhance its functionality and reduce processing times. One modified wet milling method involves much less water use and no harmful chemicals for extraction, while producing products with high purity [22].

A novel approach to overcome the issue of drying the LKF residue is to process it directly using high-temperature and high-pressure extrusion cooking that will dry, sterilise, and texturize the fibre in one high throughput and easily up-scalable process. This will give shelf-stable “extrudate pieces” that may be used as a very high fibre ingredient in foods such as breakfast cereal and muesli bars. Alternatively, the extrudate pieces can be milled to give a very high dietary fibre powder with a multitude of application possibilities [25]. The extrusion cooking techniques can improve the colour, flavour, and stability of the fibre fractions [28], as can the total dietary fibre yield [31]. Extrusion cooking, due to its high temperature, pressure, and shear force, can increase the ratio of soluble dietary fibre through the breakdown of bonds of insoluble polysaccharides, converting them into soluble fractions [32,33]. The pectin-like polymers in the cell walls of LKF thus appear as a prime target for solubilisation using extrusion cooking LKF [23]. This potential has recently been reported by Naumann et al. [15,34], who confirmed that extrusion cooking increased the solubility, water binding, and viscosity but decreased the bile acid diffusion, indicating the cholesterol-lowering potential of fibres and thus showing great potential for producing a more health-enhancing dietary fibre ingredient.

#### 3.1.2. Dry Processing: Methods, Advantages and Disadvantages

Dry processing is used to prepare fibre-enriched fractions from legumes by disintegrating seeds through the process of milling and then air classification into starch, protein, and fibre fractions [22]. For instance, pin-milling of legume seeds results in distinct populations of particles that differ in both size and density. The air classification technique is used to separate the light or fine fraction (containing mostly starches and fibres) from the coarse and relatively heavier fraction (containing mainly proteins and lipids) [35]. Air classification is repeated several times to purify further fractions [25,36]. The advantages of this dry process include: the relatively simple construction of processing plants, no wastewater production, and minimal changes in the structure and functional properties of the components [37]. Therefore, if air classification can be used as an alternative to wet processing techniques for the production of protein and fibre fractions from lupin, it has many advantages, including low capital and labour costs, a less costly effluent disposal system, and minimal sanitation.

The use of dry fractionation to separate the protein from the fibre fraction (lupin has negligible starch content), however, has some drawbacks, including the need for many repetitions of the air classification that can lead to low product recovery [22]. It has been reported that air classification is an efficient method for fractionating legumes that have starch as their main storage material, such as peas and faba beans, but less so for lupin in which the storage of non-starch polysaccharides (cell wall fibre) is more difficult to separate from the protein leading to less purity in the final fractions [25,29]. Gueguen [36] stated that the air classification process technique does not give satisfactory results for lipid-rich seeds such as lupin, wrinkled pea, chickpea, or soybean; however, *L. angustifolius* has a relatively low lipid content compared to some other lupin species and as such the lipids may not be a hindrance to its fractionation. Particle size is critical to efficient separation in air classification; therefore, by decreasing particle size through multiple passes, the purity of the targeted fraction can be improved. The fine fraction purity, such as fibre, can also be increased by lower moisture during air classification [38,39]. Sosulski and Sosulski [30] reported that when using air classification, most of the anti-nutritional factors are recovered in the fibre fraction of legumes; however, sweet varieties of *L. angustifolius* (Australian sweet lupin) are low in alkaloids, and the anti-nutritional factors associated with some other legumes (e.g., trypsin inhibitors) are absent. It is recommended that the alkaloid content of dry fractionated LKF is tested to ensure any concentration effect has not increased its levels above the maximum permitted level of 200 mg/100 g in Australia and New Zealand [40].

Recently, the dry processing method of electrostatic separation has been reviewed for the fractionation of plant materials [41]. This method is based on the different triboelectric charging properties of materials, e.g., different contact electrification when fibre and proteins in lupin are rubbed together, allowing them to be separated between electrodes. This review cited the levels of protein purification from lupin flour by a range of dry processing methods (Table 1) that can indicate the concomitant purification of the fibre fraction [41]. The reason for showing protein purity in Table 1 was that there is no published data on the dry fractionation of lupin kernels that reports the dietary fibre purity of the high fibre fraction. However, the main constituents of the lupin kernel are dietary fibre and protein, and the dry fractionation process gives two main fractions: the “high protein” fraction and the “high dietary fibre” fraction. Therefore, if a process provides the protein with a fraction of high protein purity, it follows that the dietary fibre fraction from that process gives a high dietary fibre purity. It can be seen from Table 1 that using the current technology, electrostatic separation is not superior to air classification; however, technical improvements to the electrostatic separation methods are being researched [41]. The option of pre-concentrating with dry fractionation before the final separation with wet processing is also highlighted as a potential mixed-method approach [41,42]. Figure 3 shows a schematic diagram of how electrostatic separation can be combined with more conventional methods, which may result in increased efficiencies and purities.

## 4. Composition and Techno-Functionality of LKF Food Ingredients

### 4.1. Typical Composition of LKF Food Ingredients

Table 2 presents the composition of lupin (*L. angustifolius*) kernel fibre food ingredients reported in the literature. All these examples were produced using wet processing by alkaline extraction and acid precipitation followed by the drying of the protein precipitate. Full fat lupin was used in all of the studies except that of Fechner et al. [46]. These data show the high level of purification of the dietary fibres, the residual level of protein and to a lesser extent fat, even when the kernels were not defatted. This is because most of the fat will solubilise in the highly alkaline conditions used to dissolve the protein and will not associate with the insoluble fibre residue.

### 4.2. Physicochemical Properties of LKF Food Ingredients

A wide range of physicochemical properties influences the effect of adding dietary fibre ingredients into food products. These include hydration food properties, binding of fat/oil, available surface area and porosity, fibre particle size and bulk volume, and ion exchange capacity [51]. Only a few publications have reported some of these properties for LKF.

#### 4.2.1. Colour, Odour, Flavour and Texture

Colour, odour, flavour, and texture are factors that must be considered in the application of fibre ingredients due to their impact on the sensory characteristics of foods in which they are incorporated. One aim during the manufacture of LKF is to produce a product nearly white in colour, with little odour and a neutral flavour [52]. Thus, its pale colour and low odour and flavour make LKF suitable for fibre enrichment of a wide range of foods such as dairy, baked goods, and meat products; it was dubbed an “invisible fibre” [47]. However, Stephany et al. reported non-enzymatic oxidation in LKF during storage and gave it an unacceptable odour as determined by sensory evaluation. The authors recommended preheat treatment of the lupin seed to reduce the lipoxygenase activity prior to the manufacture of the LKF. In addition, LKF has a smooth texture that makes it an excellent ingredient for fibre enrichment of food formulations [23]. In contrast, lupin kernel flour has some limitations as a fibre enrichment ingredient in foods due to its pale yellow colour and slight beany flavour [53].

#### 4.2.2. Hydration, Water Binding and Viscosity

Hydration properties of the fibre depend on the chemical and physical structures, environmental conditions of the aqueous solution, and the different processing treatments applied to extract the fibre. Hydration terms such as water binding, water holding, and water retention are used interchangeably [51]. Analytically, the water-binding capacity (WBC) of a fibre refers to its ability to bind water and hold it under centrifugal force, with high and stable water-binding properties of fibres preferable for most food product development applications [54]. However, high water binding of lupin kernel fibres can reduce the level of water available for the development of gluten in leavened bread, preventing the full formation of the visco-elastic network in the dough needed to trap gas during fermentation; therefore, for this application, lower water-binding fibres are easier to incorporate [52]. In contrast, the high water-binding capacity of LKF could result in the production of a bread with a slower staling rate through inhibiting moisture migration in the crumb and loss through the crust; however, there is no published evidence of this potentially useful phenomenon. Fibres aid in the modification of food texture through water retention. Various food processing treatments, such as extrusion or grinding, can modify hydration properties and improve functionality [51]. The water-binding capacity of LKF at 11 mL/g dry solids is high compared with other primarily insoluble fibre types, e.g., soy kernel fibre (7 mL/g dry solids), pea hull (5 mL/g dry solids), cellulose (5 mL/g dry solids), and wheat bran fibre (4 mL/d dry solids) [16]. The high water-binding capacity of the lupin kernel results from the high level of pectin-like hydrophilic non-starch polysaccharides embedded in the cell wall structure [54].

In general, the viscosity of primarily insoluble fibre ingredients such as LKF is lower than that of soluble fibre ingredients such as pectins, gums, and β-glucans. However, amongst the primarily insoluble fibres, Turnbull et al. [16] reported a higher viscosity for LKF than for soy kernel fibre, pea hull, cellulose, and wheat bran fibre. Thus, LKF has unusual hydration properties, intermediate between those of soluble and insoluble fibres ingredients [16].

#### 4.2.3. Oil Binding

Fat/oil binding is an important physicochemical property of dietary fibre food ingredients [51]. LKF can interact with oil in a food formulation, and according to McCleary and Prosky [54], this good oil-binding potential plus its water-binding capacity makes it an ideal additive for food preparation such as burgers, as well as a potential fat part-replacer in low-fat processed meat products [55].

#### 4.2.4. Consumer Acceptability of Lupin Kernel Fibre-Enriched Food Products 

In the development of fibre-enriched products, the sensory properties, including taste, are expected to be like the equivalent convention product. LKF can fulfil this desire as they have a neutral colour, odour, flavour, and smooth texture that assists in the sensory acceptability of food formulations enriched with this fibre source [48,52,54]. There are only very few published studies on the sensory acceptability of foods containing LKF. As outlined by McCleary and Prosky [54], LKF can improve baking stability without affecting taste and is an ideal choice for baking-stable fillings or toppings and coatings for fried foods due to its texturizing properties and the high and stable water-binding and good oil-binding properties. Foods incorporated with LKF, such as white bread, muffin, pasta, orange juice, and breakfast bar, have demonstrated acceptable palatability in sensory evaluation trials, though some food types of the LKF variant showed lower palatability compared to the control. In this study, the products were enriched with at least 3 g/serving of the lupin kernel fibre. No changes in overall acceptability were observed for fibre enrichment at serving levels of 4.4 and 7.3 g in bread and pasta compared to 5.5, 2.9, and 5.4 g in the muffin, the orange juice, and the breakfast bar, respectively. In all foods enriched with fibre, the flavour significantly affected the overall acceptability. This means the specific food formulation with added lupin kernel fibre requires knowledge of physicochemical properties of fibres and their interactions in the food matrices and processing-induced changes [47]. In a study by Hall et al. [56], the liking of LKF-containing foods (muesli, bread, muffin, chocolate brownie, chocolate milk drink and pasta) was evaluated after repeated consumption in a dietary setting and it was reported that the fibre addition gave no severe effects on product palatability. In contrast to LKF, lupin flour has a beany flavour that can reduce the sensory attribute of products to which it is added; this may have contributed to its slow uptake in baked products due to poor sensory quality [57].

## 5. Commercial Examples of LKF Food Ingredients and Their Use in Foods

There are very few examples of commercially available LKF food ingredients or commercial applications of it in food products. Prolupin GmbH [58] advertises that they make a dietary fibre ingredient “from the innermost parts of the seed”, suggesting it may be LKF. The fibre is described as having a smooth mouthfeel and can be used as a “roughage” source and fat substitute, such as in meat products. However, no standard product information form was available on the web page. Prolupin also markets food products containing their ingredients under the brand “Made with LUVE” [59]; however, it was not possible from the web page to determine if LKF was used in any of their products.

## 6. Evidence of Health Benefits of LKF from Human Trials

A summary of the evidence of health benefits of LKF from human (clinical) trials is presented in Table 3; each study is described in more detail and recommendations for further research are presented. There are also some cell model and animal model trials giving supporting evidence to these clinical trials, but a discussion of these is beyond the scope of this review.

### 6.1. Metabolic Syndrome Protection

Major risk factors for the development of cardiovascular disease are: (i) obesity; (ii) elevated total cholesterol (TC) and triglycerides and low-density lipoprotein cholesterol (LDL-C) concentrations; (iii) insulin resistance, which is usually indicated by high blood glucose and insulin concentrations; and (iv) high blood pressure. The presence of some or all of these risk factors is known as the metabolic syndrome; a condition afflicting up to one quarter of the world’s population. The unique properties of LKF—bioactive protein complex (Section 4.2)—could help lower these risk factors as part of a healthy diet and lifestyle.

### 6.2. Appetite and Body Weight Reduction

Foods that are highly satiating, in that they strongly reduce appetite after eating, may help in longer-term reduction in food intake and therefore assist with maintaining a healthy body weight. One critical study demonstrating the highly satiating effect of LKF in a post-meal setting was reported by Archer et al. [55]. In this study, 38 men consumed breakfast either with a full-fat sausage patty or a reduced-fat patty where some fat was replaced by lupin fibre. The participants reported that the LKF-containing breakfast gave higher (*p* < 0.05) perceptions of “fullness” in the post-meal period and lower (*p* < 0.05) total energy intake over the day than the full-fat breakfast even though it had lower total energy. The authors hypothesised that this potentially beneficial effect was due to a combination of high water-binding properties in the stomach and small intestine [16] and fermentation of the LKF to short-chain fatty acids in the colon [60].

The post-meal study by Archer et al. [55], however, does not provide direct evidence of the ability of LKF to help fight obesity. For this evidence, longer-term (over weeks or months) double-blind trials in overweight and/or obese participants are required in which the body weight of the participants on a diet with kernel fibre-containing foods is compared to one with control foods without the fibre. This study has yet to be reported for LKF but is highly recommended to gain crucial evidence of any “anti-obesity” properties of this fibre.

### 6.3. Cholesterol and Blood Pressure

High blood pressure and elevated fasting total cholesterol levels, low-density lipoprotein cholesterol and triglycerides, and low levels of high-density lipoprotein cholesterol are biomarkers of increased risk of cardiovascular disease [63]. Several studies have investigated the effect of diets containing LKF on these cardiovascular disease risk biomarkers. Hall et al. [48] found a clinically significant (*p* < 0.05) reduction in LDL-cholesterol and TC in overweight but otherwise healthy men when they consumed LKF (30 g/day) incorporated into foods compared to when they consumed the same food without the fibre addition for 28 days in a randomised cross-over study. The authors hypothesised that this beneficial effect might be due to the LKF inhibiting cholesterol re-absorption and increasing the production of short-chain fatty acids in the colon. They also noted that the residual protein in the LKF ingredient may have played a role; however, this remains unclear.

Fechner et al. [46] performed a similar study to Hall et al. [48] in participants with normal cholesterol levels but found no cholesterol-lowering effects of the LKF-containing diet. However, when they performed the study with a cohort of moderately hypercholesteremic adults, a cholesterol-lowering effect of the LKF-containing diet was found; the authors attributed this effect to increases in short-chain fatty acids butyrate and acetate found in the faeces [62].

No studies have been reported investigating the effect of LKF ingestion on blood pressure; however, several studies have shown beneficial effects on blood pressure from consumption of lupin kernel flour-containing foods in non-diabetic participants [64,65]. The authors hypothesised that the effect was due to the increased protein and fibre in the lupin flour-containing diet. In contrast, a recent similar study but in well-controlled type 2 diabetic participants found no effect of the lupin kernel flour-containing diet on blood pressure [66].

### 6.4. Protection from Type 2 Diabetes

Loss of control of blood glucose levels after a meal is a key risk factor for type 2 diabetes; therefore, the measurement of the blood glucose response of food after a meal compared with a standard and calculation of the food glycaemic index (GI) has become a popular technique for evaluating the “healthiness” of foods in terms of maintaining good blood glucose control. However, the GI is only applicable for foods with high levels of available carbohydrates (digestible starch and sugars), since the participant is required to inject a test meal containing 50 g of available carbohydrates [67]. Since the level of available carbohydrates in both lupin seed/flour and LKF is very low, their GI cannot be determined [25]. Adding lupin seed/flour of LKF to food can lower its glycemic load (GL), which is the GI multiplied by its available carbohydrate content (g/per serving) divided by 100. Inclusion of low GL foods is recommended for diets to protect against type 2 diabetes [68]. There is potential that LKF, when incorporated into foods such as wheat bread, can interact with 50 g of available carbohydrates from the wheat to lower blood glucose response, and thus the GI of the bread, through its high water-binding capacity that could slow stomach emptying and inhibit starch digestion and glucose absorption in the small intestine [27]. One study has reported the post-meal effects on the glycaemic response of food with added LKF. There was no difference in the post-meal blood glucose response in 21 healthy adults with the LKF compared to the control standard white bread [49]. The lack of effect of the LKF bread may be related to its low inclusion rate (control bread 2.7% total dietary fibre; LKF bread 8.5% TDF) in order to maintain good acceptability of the LKF bread by the participants. It is recommended that further optimisation of LKF bread manufacture is performed in order to allow a higher rate of LKF inclusion whilst maintaining palatability and to then test the glycaemic response/GI of this bread. A nutrient content labelling claim for low glycaemic index and glycaemic load can be made for foods under conditions specified by the FSANZ Food Standards Code, Schedule 4 Nutrition, health, and related claims [69].

More substantial evidence for the type 2 diabetes protective effects of LKF would be its potential to reduce fasting blood glucose and/or insulin and glycosylated haemoglobin (a marker of chronic high blood sugar levels) after consuming food containing the fibre compared to control foods without the fibre for weeks or months. One study by Hall et al. [48] reported no reduction (*p* > 0.05) in fasting glucose and insulin in 38 overweight men who consumed diets containing food with and without LKF in a cross-over design. Further studies on the effect of LKF ingestion in participants with elevated fasting blood glucose (pre-diabetic) are warranted.

### 6.5. Potential Gastrointestinal Health Benefits

The great importance of the microbiological ecology of the colon and the role of colonic bacterial fermentation of the dietary fibre for health is now fully recognised. This has recently been comprehensively reviewed with respect to the effects of whole grains and their fibre fractions consumption by Seal et al. [70]. There is good evidence from several independent studies that LKF consumption can beneficially affect microbiological (i.e., the balance of “good” probiotics and “bad” potential pathogens) and chemical markers of good bowel health and function, and thus be classified as a ‘prebiotic’ food ingredient.

Several studies have provided important evidence that LKF can act as a prebiotic ingredient and promote the growth of desirable gut bacteria while supporting digestive system function. For example, a study by Smith et al. [61] reported reduced faecal levels of Clostridia bacteria (potential pathogens) and increased levels of Bifidobacterium (beneficial probiotics) in 38 overweight but otherwise healthy men after consuming a 28-day diet incorporating food containing LKF (approx. 28 g of LKF per day) compared with a diet containing control foods with no LKF.

Improved markers of healthy bowel function (e.g., reduced transit time) and increased levels of health-protective compounds in the faeces (e.g., increased levels of the short-chain fatty acid butyrate, a substrate for healthy colonic cell development) are noted [60]. Similarly, Fechner et al. [46,62] also reported that adding LKF to the diet improved bowel function and faecal chemistry makers.

## 7. Conclusions

Dietary fibre is a major fraction of the lupin kernel, so its utilisation is critical when manufacturing food ingredients from lupin. As a food ingredient with a light colour and little flavour and aroma, LKF can act as an “invisible” fibre source to boost the fibre levels in foods, thus allowing dietary fibre nutrient content claims. Sensory evaluation trials performed with foods incorporated with LKF (at least 3 g/serving), such as white bread and pasta, demonstrated a higher acceptable palatability. In contrast, the muffin, orange juice, and breakfast bar showed lower palatability than the control. This implies that the fibre physicochemical properties and their interactions with the food systems become crucial for product formulations. In addition, the high water-binding properties of the fibre are a commercial barrier to drying it to a stable powder food ingredient. Therefore, investigations into an alternative drying approach such as extrusion cooking of the wet fibre into high fibre extrudates are suggested.

Current evidence suggests that LKF, with its unique non-starch polysaccharide structure, may have potential health benefits in the human diet, related to satiety, blood cholesterol levels and prebiotic activity; however, further independent research studies are required before health claims for this fibre can be made. LKF appears to have commercial potential as a new dietary fibre, but there are currently few, if any, commercially available LKF food ingredients.

## Figures and Tables

**Figure 1 nutrients-14-02845-f001:**
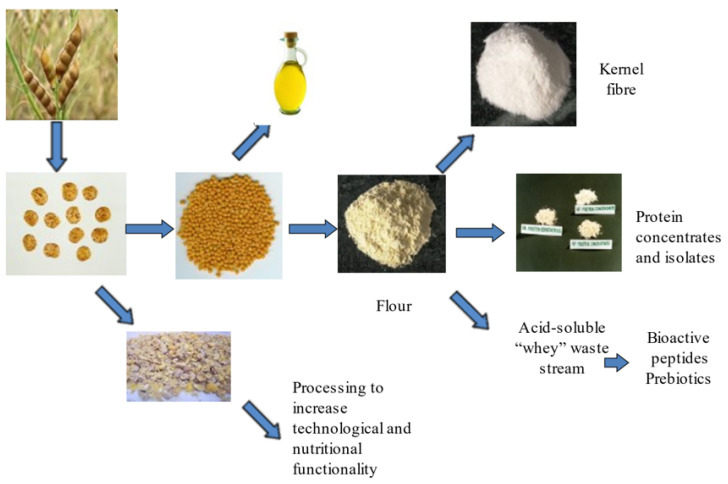
End products obtained from lupin seed processing.

**Figure 2 nutrients-14-02845-f002:**
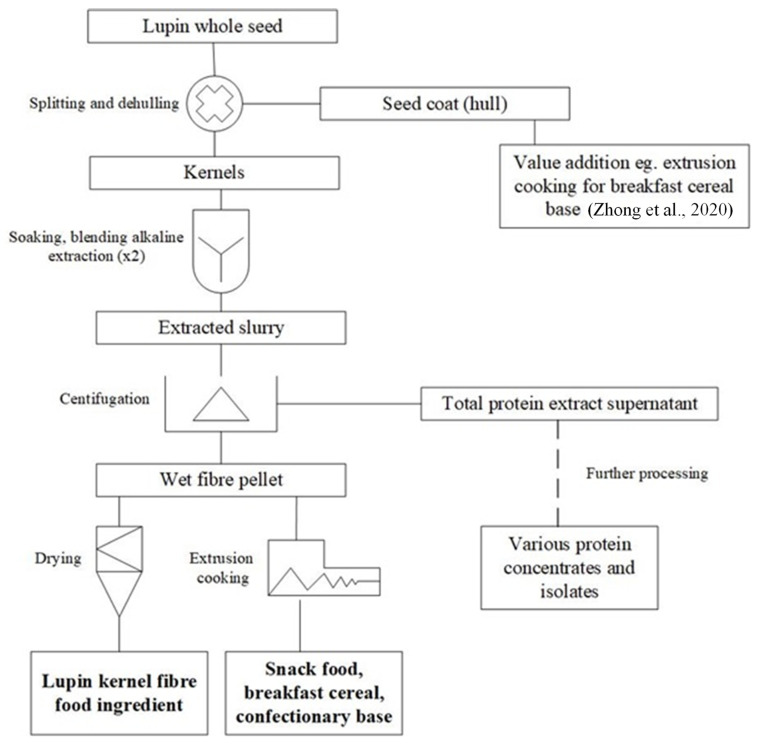
Conventional wet processing for the manufacture of LKF illustrating novel direct extrusion cooking [2].

**Figure 3 nutrients-14-02845-f003:**
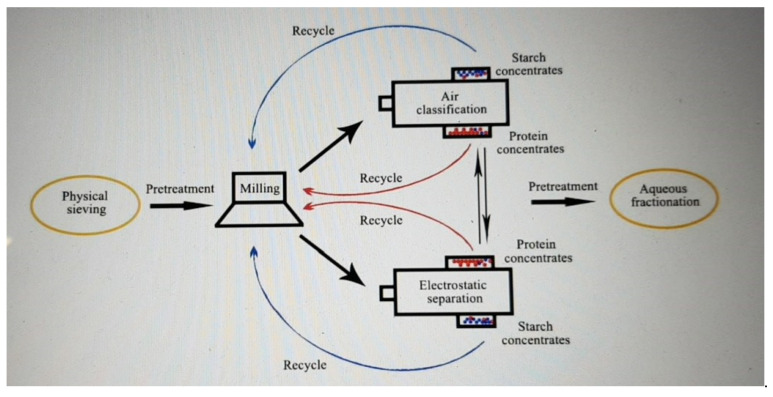
Examples of how electrostatic separation can be combined with more conventional separation methods. Reprinted with permission from [41]. 2022, Tong, L.-T.

**Table 1 nutrients-14-02845-t001:** Protein enrichment of lupin by various dry fractionation and mixed (dry + wet) methods. Adapted from [39].

Method	Protein Purity (g/100 g)		Reference
	Before	After	
Air classification	40.4	59.4	[43]
Electrostatic separation	40.5	57.3	[44]
Recycling electrostatic separation (of protein concentrate)	57.3	65.1	[44]
Air classification + electrostatic separation	45.1	59.3	[45]
Dry separation + Aqueous fractionation	53.5	>80	[42]

**Table 2 nutrients-14-02845-t002:** Proximate and dietary fibre composition of LKF food ingredients reported in the literature. These were produced by alkaline extraction and acid precipitation. ^1^ Full fat lupin used for manufacture, ^2^ defatted lupin used for manufacture.

Energy kj/100 g	Protein g/100 g	Available Carbohydrate g/100 g	Total Dietary Fibre g/100 g	Soluble Dietary Fibre g/100 g	Insoluble Dietary Fibre g/100 g	Fat g/100 g	Ash g/100 g	Reference
-	9.0	-	80.2	48.7	31.5	1.0	1.5	[46] ^1^
-	-	-	77.5	-	-	-	-	[47] ^2^
883	5.9	<0.1	88	44.8	43.2	2.1	-	[48] ^1^
-	5.7	-	77.1	8.5	68.6	2.5	1.2	[49] ^1^
-	11.1	3.7	83.3	3.7	79.7	-	1.8	[50] ^2^

**Table 3 nutrients-14-02845-t003:** A summary of the findings of clinical studies investigating effects of LKF intake on markers/physiological responses of chronic diseases.

Reference	Effect on Biomarker/Physiological Responses					
	Satiety/ Weight Loss	Blood Glucose	Blood Cholesterol	Blood Pressure	Bowel Function	Probiotic
*Short-term (post-meal) studies*						
[49]	ND	0	ND	ND	ND	ND
[55]	+ve	ND	ND	ND	ND	ND
*Longer term (dietary intervention) studies*						
[48]	ND	0	+ve	ND	ND	ND
[60]	ND	ND	ND	ND	+ve	ND
[61]	ND	ND	ND	ND	ND	+ve
[46]	ND	ND	0	ND	+ve	ND
[62]	ND	ND	+ve	ND	+ve	ND

+ve the lupin treatment gave significantly improved levels of the biomarker/physiological response compared to the control (non-lupin) treatment; 0 no difference in the levels of the biomarker between the lupin and the control treatment; ND biomarker not assessed, or experimental design not valid for comparison.

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
