# Peer review of "Lupin Kernel Fibre: Nutritional Composition, Processing Methods, Physicochemical Properties, Consumer Acceptability and Health Effects of Its Enriched Products"

_nutrients, 2022, doi:10.3390/nu14142845_

Round 1

Reviewer 1 Report

The review presents an overall literature study on the potential benefits of lupin kernel fibers as food ingredients. Thereby, it focusses on the composition, processing as well as the assessment of functional and bioactive properties.

However, the authors are encouraged to thouroughly revice the whole manuscript as it is not well arranged. Particularly, several of the sections are misleading as the authors often describe the impact on protein purity rather than the impact of processing on lupin kernel fibers. The following major issues should be thouroughly revised prior to the decision of potential acceptance:

The section on the general properties of dietary fibers and their benefits as some of the statements are not totally in line with existing literature. In particular, it is described in detail that both insoluble and soluble dietary fibers affect glucose metabolism, but with different mode of actions. This aspect showed be emphasized in the manuscript.

In section 2.1 it is stated the "The remainder of this literature review will now focus on LKF". I think this is completely wrong as the authors focus further on LKF, but not on hull fibers.

There are two sections 2.1 with the same wording and there are several repetitive sections, which are sometimes even not formulated in complete sentences. Furthermore, a section on the composition of LKF is missing. How could that happen? Did the author not proof-read their mansucript prior to submission? This is the key section of the review and therefore, could not be judged by the reviewers due to lack of inclusion.

Figure 2 should be revised as the author state that no investigations on the properties of extruded fibers remain. If so, the Figure is misleading and extrusion might be an option, which is not yet investigated in detail. In addition, the review describes the impact of extraction on composition and functionality of LKF, but no data is displayed. This would increase the significancy of this review tremendously.

Why did the author add Table 1 with data on protein purity rather than data on LKF? The section seems to focus on protein obtained by dry fractionation, but the main topic of this review is dietary fibers. This should be improved. Similarly, Figure 3 is also misleading. Therefore, it is highly recommended to improve the alignment of the review and focus on the topic stated in the title.

Section 4.1. some recent publications are missing such as the publications from the Wageningen group or from Naumann et al. Those researchers also investigated blue lupin kernel fibers in detail. Furthermore, table 2 should be revised as several compositional characteristics are only available by one publication such as the carbohydrate content. Therefore, it is recommended to shorten the table.

Similarly, table 5 is inconclusive. Maybe it is easier to display those findings in text. Also, some investigation on lupin kernel fiber and there effect in humans is missing such as the ones published by Anna Arnoldi and others.

The characteristics of aroma and flavour of lupin kernel fibers are only stated, but not properly underpinned by literature. The authors are referred to the investigations of Stephany et al.

The conclusions drown by the authors are very weak, particularly the hints that lupin kernel fibers are a valuable source for increasing satiety and other bioactive properties as literature data displayed in the manuscript partially contradicts the conclusion of the authors. I would recommend that the authors draw their conclusions in a more conservative way.

The following minor issues should be considered during revision:

Figure caption1: Please review according to English grammar

db. I guess this abbreviation means on dry basis. Maybe you should mention this at the first reference?

The following two sentences should be properly interlinked. "In lupin seeds, the dietary fibre is located in the protective seed coat (hull) at 80 – 87 g/100 g db (Zhong et al., 2020). Also, the kernel (endo-sperm) is unique amongst legumes as it contains high dietary levels of approx. 40 g/100 g db (Villarino et al., 2014)."

Please inlcude the citation correctly in the following sentence: "There are many excellent and comprehensive reviews on the health benefits of dietary fibre for humans eg. (Anderson et al., 2009)".

The following sentence should be revised: high temperature high pressure extrusion to high temperature high pressure extrusion cooking

The citation should be included correctly in the following sentence and the correct citation should be inlcuded which is Naumann et al 2021: "This potential has recently been reported by (Naumann et al., 2019) who confirmed that ex-trusion cooking increased the solubility, water binding and viscosity and thus has great potential for producing a more health-enhancing dietary fibre ingredient."

Author Response

The authors wish to thank the reviewers for their detailed feedback and suggestion. We have carefully responded to these suggestions as detailed below. Modifications to the text are given in red in the revised manuscript.

Reviewer 2 Report

Reviewer comments

In this narrative review, Malekipoor et al. reviewed the nutritional composition, effect pf processing methods and techno-functional properties of lupin kernel fibre. The consumer acceptability and potential effects of lupin kernel fibre-enriched food products was also reviewed. Please, find below my comments for your response.

Title: The authors should consider revising the title “Lupin kernel fibre: a new bioactive food ingredient?” This is because the title does not solely match the objectives of the review. There was no section on the “Bioactive composition and antioxidant properties of the lupin kernel fibre” for example. Consequently, I suggest the authors revise it to “Lupin kernel fibre: Nutritional composition, processing methods, physicochemical properties, consumer acceptability and health effects of its enriched products”.

Abstract

Primarily, reviews are to highlight gaps in literature for further works to be carried out on the identified gaps. Consequently, I suggest the authors consider these points below for their inclusion in the Abstract as a form of “Recommendation”. The authors should recommend that, “From the review, it is evident that more processing non-thermal methods should be explored in lupin kernel fibre processing and its effects on the physicochemical properties investigated.” Also, from the health effects, it was evident that the clinical trials are few. Thus, there is the need to for long term clinical trials investigating the effect of lupin kernel fibre-enriched product intake on health outcomes.

Keywords: The authors should include “sensory evaluation” OR “consumer acceptability”

Introduction

The title of figure one should be revised. You could consider “End-product obtained from Lupin seed processing”.

Page 2: Second paragraph, In this statement “Also, the kernel (endo-sperm) is unique amongst legumes as it contains high dietary levels of approx. 40 g/100 g db (Villarino et al., 2014)”, the authors should introduce “fibre” before the “levels”.

Page 3: In this sentence “There are many excellent and comprehensive reviews on the health benefits of dietary fibre for humans eg. (Anderson et al., 2009)”, the authors use “many reviews” yet they cite just one paper. They should consider adding more. If it is just that one review paper, they should revise the sentence and replace “many” with “a”.

Page 3: The authors should provide a reference for these statements “In general, insoluble fibres can provide health benefits to the human gastroin-testinal tract mainly through their bulking ability that can stimulate healthy bowel movement. However, soluble fibres are generally considered to have more beneficial metabolic effects than those from insoluble fibres. Their viscosity can help modulate nutrient digestion and absorption (eg. blood glucose control). On the other hand, their fermentability can both (a) assist in maintaining a healthy gut microflora and (b) produce a range of metabolically beneficial fermentation products such as short-chain fatty acids that assist with cholesterol control and provide a substrate for healthy colon cell development.”

Page 3: In this sentence “Lupin seed hull has been used as an ingredient in human food such as in high fibre bread and in meat products as a bulking agent (Zhong et al., 2018), however most of it undergoes little if any value addition”, the authors should clarify what “……………however most of it undergoes little if any value addition”.

Page 3: The authors should revise this sentence “The remainder of this literature review will now focus on LKF. significant extent in the human colon; they primarily consist of cel-lulose, hemicelluloses and lignin. In contrast, soluble fibres are highly water-soluble and highly fermentable in the colon and include pectins, gums and mucilages (Dhingra et al., 2012).”

Under the section title “2.1. Dietary fibre of lupin hull”

This statement “There are many excellent and comprehensive reviews on the health benefits of dietary fibre for humans eg. (Anderson et al., 2009). In general, insoluble fibres can provide health benefits to the human gastroin-testinal tract mainly through their bulking ability that can stimulate healthy bowel movement. However, soluble fibres are generally considered to have more beneficial metabolic effects than those from insoluble fibres. Their viscosity can help modulate nutrient digestion and absorption (eg. blood glucose control). On the other hand, their fermentability can both (a) assist in maintaining a healthy gut microflora and (b) produce a range of metabolically beneficial fermentation products such as short-chain fatty acids that assist with cholesterol control and provide a substrate for healthy colon cell development”, is repeated above.

In Page 4, there is a repetition of the section title “2.1. Dietary fibre of lupin hull” with its content.

Page 4: The statement “The remainder of this literature review will now focus on LKF. that mainly consist of non-starch polysaccharides and raffinose family oligosaccharides such as raffinose, stachyose and verbascose (Evans et al., 1993; Trugo et al., 2003)”, is not clear. The authors should revise it.

Page 4: In this section title “3.1. Processing approaches”, The authors could consider dividing the section into two using “Thermal and Non-Thermal processing techniques that could be adopted for lupin kernel fibre processing”.

Page 5: In this sentence “An example of a wet pro-cessing scenario to produce highly dietary-fibre enriched LKF is……”. The authors should remove “highly”.

Page 8: This sentence “However, these used dried LKF and therefore there remains no studies reporting the effect of direct extrusion of the wet fibre paste on the fibre properties”, should be revised.

Page 7: In this sentence “(Gueguen, 1983) stated………..”, the authors should revise it to “Gueguen (1983)…..

Page 7: The authors should revise “(Sosulski & Sosulski, 1986)…….” to “Sosulski & Sosulski (1986)….”

Page 9: Under this section “Typical composition of LKF food ingredients”, In this sentence “These data show the high level of purifying of the dietary fibres,……”, the authors should replace “purifying” with “purification”

Page 9: I don’t get why the authors have put Table 2 under the section title “Physicochemical properties of LKF food ingredients”. The authors should create a section title on “The nutritional composition of Lupin kernel fibre and place this Table there.” Or they could also place it under the sections on the processing methods discussed in the preceding sections and capture it as “Effect of processing methods on the nutritional composition of lupin kernel fibre”. This is because “alkaline extraction and acid precipitation” were employed.

Page 10: This section title “4.2.2. Hydration, water-binding and viscosity” should be revised to “4.2.2. Hydration, water-binding and viscosity properties of lupin kernel flour”.

Page 10: Line 8-9, the authors should introduce a statement about the effect of high water retention of fibre in bakery products before introducing the statement. They could mention that the high water saturation could dilute the gluten and impair the visco-elastic network that will be needed for to trap carbon dioxide gas that will be produced after fermentation.

The authors should also that, the rich pectin composition of the lupin kernel fibre and for that matter making it have a high water holding property could results in the production of bread with softer crumb compared to wheat bran that usually results in bread with harder crumb due to its poor water holding capacity. Consumers prefer bread with softer crumb as it could be easier to chew.  The authors could consider discussing the potential “Effect of the physicochemical properties of the lupin kernel fibre” on the “Textural properties of bakery products especially bread” as it is one of the most commonly consumed staple foods.

The authors could also highlight the effect of applying some emerging pre-treatment methods including “Pulse-electric field method for example on the functionality of the lupin kernel fibre”.

Page 10: Line 21: Please correct this “(Turnbull et al., 2005)” to “Turnbull et al. (2005)”.

Page 10: Line 27: This sentence “LKF can interact with oil in a food formulation and ac-27 cording to (McCleary & Prosky, 2008)”, is not complete.

Page 10: Line 31: Revise “Consumer panel acceptability” to “Sensory evaluation and consumer acceptability of lupin kernel fibre-enriched products”. The authors should give more insight on especially these three organoleptic attributes “Colour or Appearance, Taste and Texture” of the lupin kernel fibre-enriched products. They can also compare how the “Overall acceptability” of the lupin kernel fibre-enriched products compare with their “Control”. In fact, it will even be better if the authors state the level of lupin kernel fibre substitution which is more acceptable in comparison to the control sample

Conclusion

The authors should indicate the effect of lupin kernel fibre enrichment on the sensory profile of its enriched products.

General comments:

A major challenge with this manuscript is the authors failing to use continuous line numbering which is required of all Nutrient manuscripts. It makes it difficult for the reviewer to make references to the authors in regards to effecting some corrections.

The authors have used the “APA referencing style” for this manuscript. However, this is not the standard reference style for Nutrients journal and the authors must thus consider changing it.

Author Response

(The authors gave the same response as above.)

Round 2

Reviewer 2 Report

Thank you for undertaking the revisions. I can see that the quality of the manuscript has subsequently improved. Please, find below some minor revisions for your response.

Line 74: There should be a full stop after [9]

Line 113: Kindly, revise “(“glacturonan”)”  to “galacturan”

Line 120: kindly, revise the sentence

Line 304: kindly remove “panel”. The authors should kindly consider making it “Consumer acceptability of lupin kernel fibre-enriched food products”.

Table 5.The authors should revise those section titles as  “Bowel function” and “Prebiotic” are not biomarkers.

Author Response

The authors wish to thank the reviewer for further feedback and suggestions. We have carefully responded to these suggestions as detailed in the revised version. Modifications to the text are given in track changes in the revised manuscript.
